# Fire-derived phosphorus fertilization of African tropical forests

Marijn Bauters[1,2 ✉], Travis W. Drake[3], Sasha Wagner[4], Simon Baumgartner[1,5], Isaac A. Makelele[1,6], Samuel Bodé[1], Kris Verheyen[2], Hans Verbeeck[2], Corneille Ewango[7], Landry Cizungu[8], Kristof Van Oost[5] & Pascal Boeckx[1]

Central African tropical forests face increasing anthropogenic pressures, particularly in the form of deforestation and land-use conversion to agriculture. The long-term effects of this transformation of pristine forests to fallow-based agroecosystems and secondary forests on biogeochemical cycles that drive forest functioning are poorly understood. Here, we show that biomass burning on the African continent results in high phosphorus (P) deposition on an equatorial forest via fire-derived atmospheric emissions. Furthermore, we show that deposition loads increase with forest regrowth age, likely due to increasing canopy complexity, ranging from 0.4 kg P ha$^{-1}$ yr$^{-1}$ on agricultural fields to 3.1 kg P ha$^{-1}$ yr$^{-1}$ on old secondary forests. In forest systems, canopy wash-off of dry P deposition increases with rainfall amount, highlighting how tropical forest canopies act as dynamic reservoirs for enhanced addition of this essential plant nutrient. Overall, the observed P deposition load at the study site is substantial and demonstrates the importance of canopy trapping as a pathway for nutrient input into forest ecosystems.

[1] Department of Green Chemistry and Technology, Ghent University, Ghent, Belgium. [2] Department of Environment, Ghent University, Ghent, Belgium. [3] Department of Environmental Systems Science, Swiss Federal Institute of Technology, ETH Zurich, Zurich, Switzerland. [4] Department of Earth and Environmental Sciences, Rensselaer Polytechnic Institute, Troy, NY, USA. [5] Earth and Life Institute, UCLouvain, Louvain-la-Neuve, Belgium. [6] Department of Biology, Université Officielle de Bukavu, Bukavu, Democratic Republic of the Congo. [7] Faculty of Renewable Natural Resources Management, University of Kisangani, Kisangani, Democratic Republic of the Congo. [8] Soil Science Laboratory, Faculty of Agronomy, Université Catholique de Bukavu, Bukavu, Democratic Republic of the Congo. ✉email: Marijn.Bauters@UGent.be

Tropical forests store a substantial amount of the Earth's terrestrial carbon and host a large share of global biodiversity. Although nitrogen (N) is a potentially limiting plant nutrient early-on in tropical forest succession[1,2], it is a virtually infinite resource that can be fixed biologically from air[3–5]. In contrast, phosphorus (P) has been identified to be the main limiting nutrient in many old-growth tropical forests rooted in strongly weathered soils[6,7]. As such, these forest ecosystems are dependent upon trace inputs of P and cations from the atmosphere (e.g., as dust or biomass-derived emissions) rather than inputs from soil weathering[8,9]. In light of the dependency of tropical forests on these atmospheric P inputs, it is hypothesized that forest canopies may function as a 'trap' for atmospheric constituents[10,11]. As they regrow, secondary forests exhibit increased canopy complexity (e.g., surface area or roughness), which enables more efficient trapping of atmospheric constituents[12]. Furthermore, repeated deforestation can reduce atmospheric P inputs to a level that induces a long-term negative ecosystem P balance[13]. As such, the loss of this canopy trap through repeated clearing might also reduce cumulative P inputs to an extent where some forests completely lose the capacity to naturally recover from disturbances[14]. Understanding the biogeochemical and physical interactions of canopy trapping and atmospheric P deposition along secondary forest succession is vital for tropical landscapes, which face increasing anthropogenic pressures. These effects are particularly relevant in Africa, where secondary forests are expected to increasingly dominate the landscape as a result of ongoing slash-and-burn agricultural practices[15,16]. The African continent represents nearly 65% of the global burnt area annually[17], mainly via a high incidence of grassland and savanna fires[18]. This biomass burning leads to high fire-derived nitrogen deposition loads on central African forests[19], but the deposition of potentially limiting nutrients, such as P, and the interaction with canopy trapping along forest regrowth trajectories, is unknown.

We quantified atmospheric P deposition along a forest successional gradient in the central Congo basin to estimate the magnitude of biomass burning-derived P deposition and to assess the importance of canopy trapping in regulating the delivery of P to the forest floor. For this, we set up permanent precipitation collectors in 40 × 40 m monitoring plots along a successional gradient comprising five stages: agriculture, 5-year-old, 12-year-old, 20-year-old, and 60-year-old secondary forests, near Kisangani. Per successional stage, we set up three plots, and each plot was equipped with eight rainfall collectors, which were sampled weekly. Nutrient loads in open field and throughfall was calculated by multiplying the rainfall volume with the total nutrient concentration per plot, per week. We used open-field precipitation as a 'wet' deposition endmember, and calculated net throughfall loads to quantify dry deposition. To date, it has proven difficult to quantify net nutrient deposition in forest ecosystems, as canopy uptake/leaching effects are difficult to separate from dry deposition effects in throughfall samples. Here, we used dissolved black carbon (DBC; measured as benzenepolycarboxylic acid molecular markers[20]) as a tracer for biomass burning-derived aerosols in throughfall samples. The burning of biomass releases nutrients and produces black carbon, a heterogeneous mixture of charcoal, soot, and other thermally altered forms of organic carbon[21]. We related DBC deposition to P deposition (both wet and dry) and then compared results to canopy complexity in forests along a successional gradient (varying in age from 5 to 60 years since agricultural abandonment). Note that wet deposition in this study is not wet-only deposition, but the total deposition in the open-field collectors, which were not wet-only collectors, but continuously exposed.

## Results and discussion

**High P deposition on central African forests is sourced from biomass burning.** Our data show that open-field wet deposition at our site in central Africa (Fig. S1) is $0.43 \pm 0.09$ kg P ha$^{-1}$ yr$^{-1}$, while 60-year-old forests are subjected to an average total deposition load (i.e., wet and dry deposition) of $3.1 \pm 1.4$ kg P ha$^{-1}$ yr$^{-1}$. These measured P loads are substantially higher than what has been previously estimated for the region ($0.8$–$1.0$ kg P ha$^{-1}$ yr$^{-1}$)[22], but similar to total P deposition measured on the shores of Lake Victoria ($1.8$–$2.7$ kg P ha$^{-1}$ yr$^{-1}$)[23]. We analyzed DBC in pooled samples (see 'Methods') to determine whether deposited P measured in throughfall (Fig. 1) was indeed related to biomass burning. Since dry P deposition scales linearly with DBC dry deposition along the forest chronosequence (Fig. 1), it indicates that (1) the excess P input measured as net throughfall under forest canopies is fire-derived, and (2) net P throughfall loads (i.e., the throughfall P minus the open-field P deposition) are derived entirely from biomass burning, rather than canopy leaching. A pyrogenic source for throughfall P is consistent with previous studies that suggest an estimated >50% of global atmospheric P is derived from combustion[24]. Africa is also known to be a hotspot for biomass burning and constitutes over half of the global carbon emissions from fire[18], with the fire incidence being most abundant in the savanna zones (Fig. S2). Additionally, localized P emissions from biomass burning have been shown to be[25] up to 20 kg P burnt ha$^{-1}$ owing to the rather high P emission factors from tropical forest and savanna fires[26].

**Canopy trapping of atmospheric nutrients increases with forest age and canopy complexity.** Annual P deposition in the 60-year-old forest ($3.1 \pm 1.4$ kg P ha$^{-1}$ yr$^{-1}$; Fig. 2) was found to be twice that of annual P deposition in 5-, 12-, and 20-year-old forests, which exhibited similar values ($1.6 \pm 0.7$ kg P ha$^{-1}$ yr$^{-1}$, $1.5 \pm 0.3$ kg P ha$^{-1}$ yr$^{-1}$, and $1.1 \pm 0.2$ kg P ha$^{-1}$ yr$^{-1}$, respectively) which in turn relate well to simulated estimates of annual P deposition for the area[24]. To further examine the role of forest age and structure on P deposition, we quantified canopy complexity, using high-resolution 3D point clouds derived from unoccupied aerial vehicle (UAV) imagery, as canopy 'roughness' (Fig. S3). Canopy roughness is linearly related to the dry deposition velocity of aerosol particles[27,28]. Canopy complexity encompasses both the structural variability of the canopy, as well as the vegetation height. Additionally, we used canopy rugosity, the standard deviation of vertical variability in the upper canopy layer, as a simple structural metric related to canopy complexity. As hypothesized by Powers et al.[12], our data suggest that more complex canopies, that is, with higher canopy roughness and rugosity, trap more P via dry deposition (Fig. 2). This highlights canopy trapping as a potentially important ecosystem-level mechanism for nutrient acquisition. Our data also suggest that forests trap nutrient-rich aerosols with increasing efficiency as they recover from disturbance. The first 20 years of forest succession exhibited a plateau in the structural variability of the canopy (see λ in Table S1), which is mirrored by consistent P dry deposition loads. However, both canopy complexity and P dry deposition loads increased in 60-year-old forests. The observed variability in canopy trapping across the successional gradient, which lead to three-fold differences in P input to the forest floor, might have important ramifications for the nutrient cycles of central African forests. Indeed, reduced canopy cover also reduces the local atmospheric inputs of nutrients to the soil. Thus, long-term absence of a canopy might thereby enhance a long-term negative ecosystem nutrient balance and constrain future forest recovery, as described by Lawrence and colleagues[13].

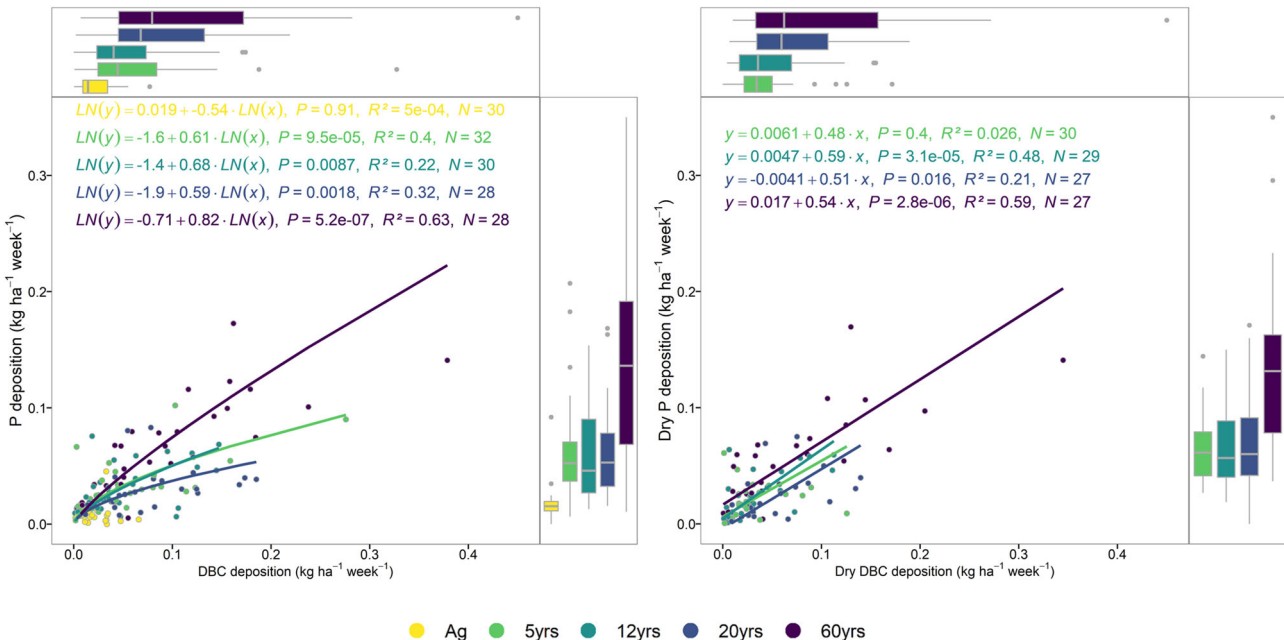

**Fig. 1 Weekly total phosphorus deposition versus dissolved black carbon deposition (DBC deposition) and dry P deposition versus dry DBC deposition along successional stages of central African forests: agricultural field (Ag), 5- (5 yrs), 12- (12 yrs), 20- (20 yrs), and 60-year-old forest (60 yrs).** Boxplots show the median, the 25th, and 75th percentile and the 1.5 interquartile range spread. Points represent individual, weekly, measurements of composite samples ($n = 8$) per plot for each successional stage.

**Forest canopies as temporary reservoirs for P deposition.** We used simulated black carbon surface mass concentration (BCSMC) from the MERRA2 model to assess whether temporal P deposition patterns could be explained by temporal variations of black carbon in the atmosphere. However, we did not find a significant relationship between the rate of P deposition and the BCSMC over the site. Instead, weekly P throughfall amounts—or post-dry-deposition P wash-off—were shown to be correlated with weekly rainfall on a log–log basis across each site (back-transformed relation shown in Fig. 3 and Table S4). While the BCSMC temporal pattern, which is consistent with the aerosol optical depth at 550 nm via MODIS' Terra satellite, shows that the intensity of dry deposition varies over the year (Fig. 4), significant correlations between P throughfall and rainfall suggest that wash-off of dry deposition is controlled by the amount of rainfall available to wash particles from the canopy rather than the timing of their arrival on the canopy. In other words, P input to lowland forest floor is a transport-limited process (Fig. 3). As such, canopies act as dynamic reservoirs where dry deposition accumulates over time, replenishing the forest floor with an amount of deposited P in proportion to the amount of rainfall. In fact, our data show that this is not only valid on a rainfall event basis, but also that canopies can act as year-round stores for nutrients.

Overall, we found that central African forests receive P deposition loads that are substantially higher than expected from model simulations and are sourced by biomass burning on the continent. Additionally, canopy complexity is likely a major driver of nutrient inputs, where older and more complex canopies more efficiently trap atmospheric constituents. This dry deposition trapping increases the bulk 'open-field' deposition from 0.4 to 3.1 kg ha$^{-1}$ yr$^{-1}$ in old secondary forests, which means that forests are about eight times more effective at capturing atmospheric P compared to non-forest ecosystems. Local P deposition is a spatially heterogeneous process controlled by both on-site biotic factors (e.g., canopy complexity), as well as regional-scale determinants (e.g., continental fire regimes). Our estimate of

total P deposition in older forests of the Congo basin is three times the amount that would be needed to sustain tropical forest growth in steady state across the tropics (1.1 kg ha$^{-1}$ yr$^{-1}$)[29]. Hence, the magnitude of these deposition loads raises the question whether the widespread assertion of P limitation in old-growth central African forests holds[30,31]. At present, biomass burning on the African continent is slightly lower than it has been over the last millennia[32], which suggests that the magnitude of annual P deposition has remained high over the same duration. However, given the likely increase in biomass burning related to anthropogenic activities, P deposition is likely to increase in the coming decades.

## Methods

**Study site.** The study was carried out in post-agriculture forests at different growth stages near the forest reserve of Yoko (N00°17′; E25°18′; mean elevation 435 m a.s.l.), situated between 29 and 39 km south east of Kisangani, in the Democratic Republic of the Congo. We set up 15 (40 × 40 m) plots, set out in triplicate along five successional stages (15 plots): agriculture and 5, 12, 20, 60 years old secondary forest (respectively, 5 yrs, 12 yrs, 20 yrs, 60 yrs). Additionally, soils were also characterized in three agricultural plots (Ag). We interviewed owners, farmers, and local experts to determine the time-since-disturbance of all plots. Tree height measurements were recorded at the plot level for 20% of individuals of each diameter class. The climax vegetation in the region is classified as semi-deciduous tropical. Climate falls within the Af-type following the Köppen-Geiger classification[33]. Annual rainfall ranges from 1418 to 1915 mm with mean monthly temperatures varying from 23.7 to 26.2 °C. Throughout the year, the region is marked by a long and a short rainy season interrupted by two small dry seasons December–January and June–August. Soils in the region are highly weathered Oxisols, being poor in nutrients, with low pH and dominated by sandy texture.

**Sampling and sample analysis.** Throughfall and bulk precipitation was collected weekly using polyethylene (PE) funnels supported by a wooden pole of 1.5 m height to which a PE tube was attached and draining into 5 L PE container. A nylon mesh was placed in the neck of the funnel to avoid contamination by large particles. The container was buried in the soil and covered by leaves to avoid the growth of algae and to keep the samples cool. We installed eight throughfall collectors in each plot as two rows of four collectors, with approximately 8 m distance between all collectors. On every sampling occasion, the water volume in each collector was measured in the field, and recipients, funnels and mesh were replaced, rinsed with distilled water. A volume-weighted composite sample of the devices per plot was made. All samples were stored in a freezer immediately and sent in batch to Belgium for chemical analysis. The volume-weighted composite

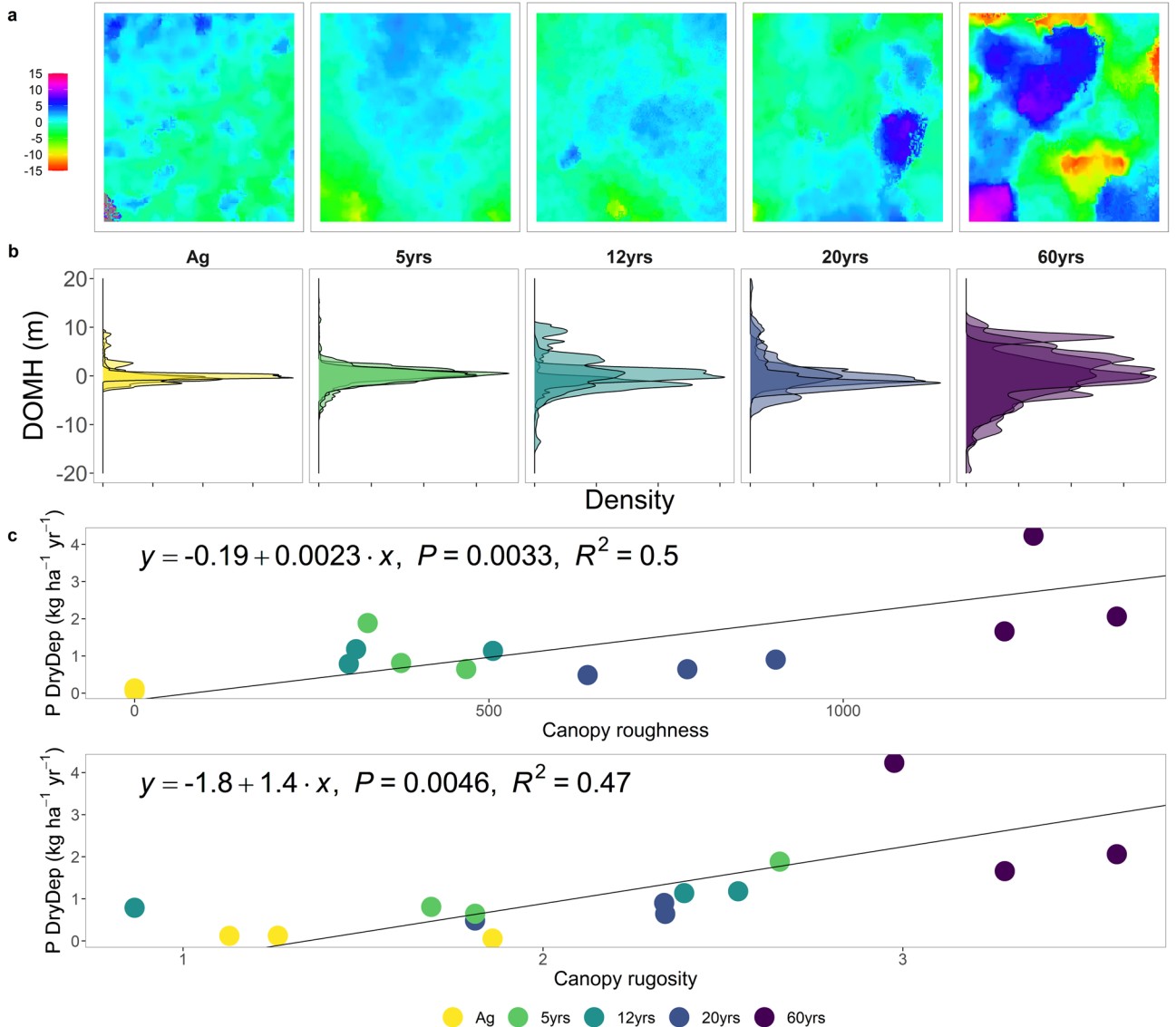

**Fig. 2 Canopy complexity and annual phosphorus dry deposition along successional forest stages: agricultural field (Ag), 5- (5 yrs), 12- (12 yrs), 20-(20 yrs), and 60-year-old forest (60 yrs). a** Canopy height models show the deviation in canopy height (m) from the mean of a 40 × 40 m plot per successional stage. **b** Density distributions for three plots per successional stage displaying the deviation of mean height (DOMH, m) from the plot's mean. **c** Canopy roughness and rugosity, based on unoccupied aerial vehicle-derived structure for motion and vegetation height, versus the annual P dry deposition. Points in (**c**) represent the plot-level measurements for annual P deposition and canopy roughness and rugosity.

samples were first filtered using a nylon membrane filter of 0.45 μm before freezing. Total phosphorus was measured by inductively coupled plasma atomic emission spectroscopy (ICP AES, IRIS interpid II XSP, Thermo Scientific, USA). Although we acknowledge the potential for microbial activity in the collectors during a 1-week, dark, in situ storage of the samples, the use of total phosphorus concentration and lack of algal growth allow for complete phosphorus recovery.

Following analysis, the samples from the replicate field sites per forest stage were pooled into 'weekly' forest-type samples, and these were subsequently analyzed for dissolved black carbon (DBC). In short, the pooled water samples were acidified to pH 2 and analyzed for dissolved organic carbon (DOC) concentration via high-temperature catalytic oxidation on Shimadzu TOC-L total organic carbon analyzer following established methodology[34]. DOC was isolated from the water samples by solid phase extraction (SPE) following Dittmar et al.[35]. Briefly, SPE cartridges (Varian Bond Elut PPL, 1 g, 6 mL) were conditioned sequentially with methanol, ultrapure water, and ultrapure water acidified to pH 2 using concentrated HCl, then passed through the SPE cartridges by gravity. SPE cartridges were dried under a stream of high-purity $N_2$ gas. DOC was eluted from the SPE cartridge with methanol (SPE-DOC) and stored at −20 °C until further analysis. DBC was quantified using the benzenepolycarboxylic acid (BPCA) method as detailed in Wagner et al.[20]. The BPCA approach to quantifying DBC involves chemothermal oxidation of condensed aromatic DOC compounds to benzenehexacarboxylic acid (B6CA) and benzenepentacarboxylic acid (B5CA)

products. The B6CA and B5CA oxidation products are robustly measured and derive exclusively from pyrogenic sources[36]. Condensed aromatic DBC, as measured using the BPCA method, is ubiquitous in aquatic environments globally[21,37–39]. DBC has also been quantified in throughfall and stemflow in longleaf pine forests that undergo regular prescribed burning[40]. Therefore, we use the BPCA method as a proxy for carbon inputs from biomass burning in the current study. To analyze our samples for BPCAs, aliquots of SPE-DOC (~0.5 mg C equivalents) were combined with concentrated $HNO_3$ in flame-sealed glass ampoules and heated to 160 °C for 6 h. The resultant BPCA-containing residue was dried and re-dissolved in mobile phase for subsequent analysis. Individual BPCAs were separated and quantified using an HPLC system (UltiMate 3000, Thermo Fisher, Germany) (CV < 5%). Sample DBC concentrations were calculated using the established power relationship between DBC (μM C) and the sum of B6CA and B5CA (nM-BPCA) using the equation from Stubbins et al.[41]: $[DBC] = 0.0891 \times ([B6CA + B5CA])^{0.9175}$; $n = 351$, $R = 0.998$, $p < 0.0001$. We report the concentrations of B5CA and B6CA with DBC to facilitate comparison with other datasets (Table S2).

**Canopy 3D model.** UAV surveys were carried out in Yoko on February 9, 2020. The Yoko survey consisted of five flights due to the dispersed locations of the 15 inventory plots and in total covered an area of 302 ha. Two UAV platforms were used in the surveys: (i) a consumer-grade DJI Mavic 2 Pro. This UAV was

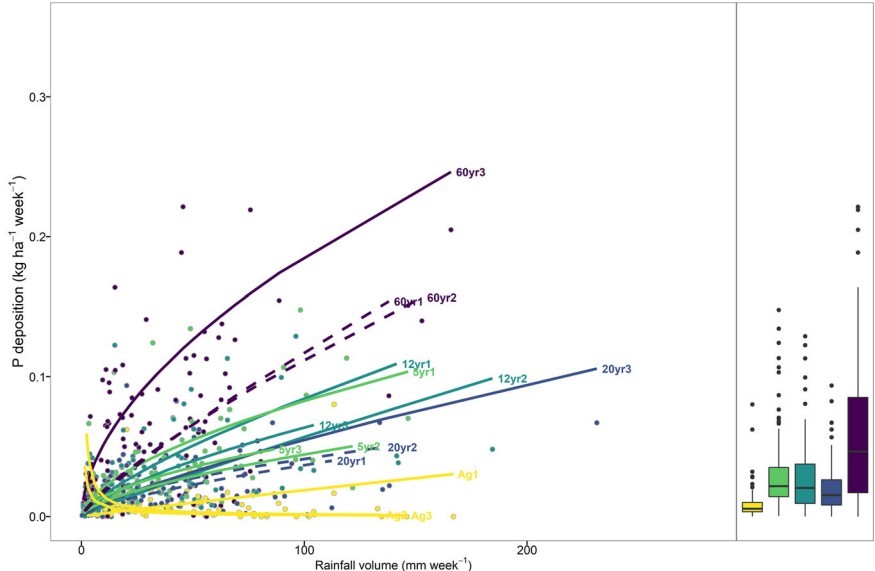

**Fig. 3 Weekly rainfall amount (open-field rainfall volume for agricultural fields, throughfall volumes for the forest stages) versus weekly phosphorus (P) deposition, in agricultural field (Ag), 5- (5 yrs), 12- (12 yrs), 20- (20 yrs), and 60-year-old forest (60 yrs).** Solid lines show significant log–log fits via standardized major axis regression, while dashed lines indicate insignificant regressions (P-value > 0.1). Boxplots show the median, the 25th and 75th percentile and the 1.5 interquartile range spread for the weekly P deposition loads. Points in represent the individual weekly measurements of composite samples (n = 8) per plot for rainfall volume and P deposition.

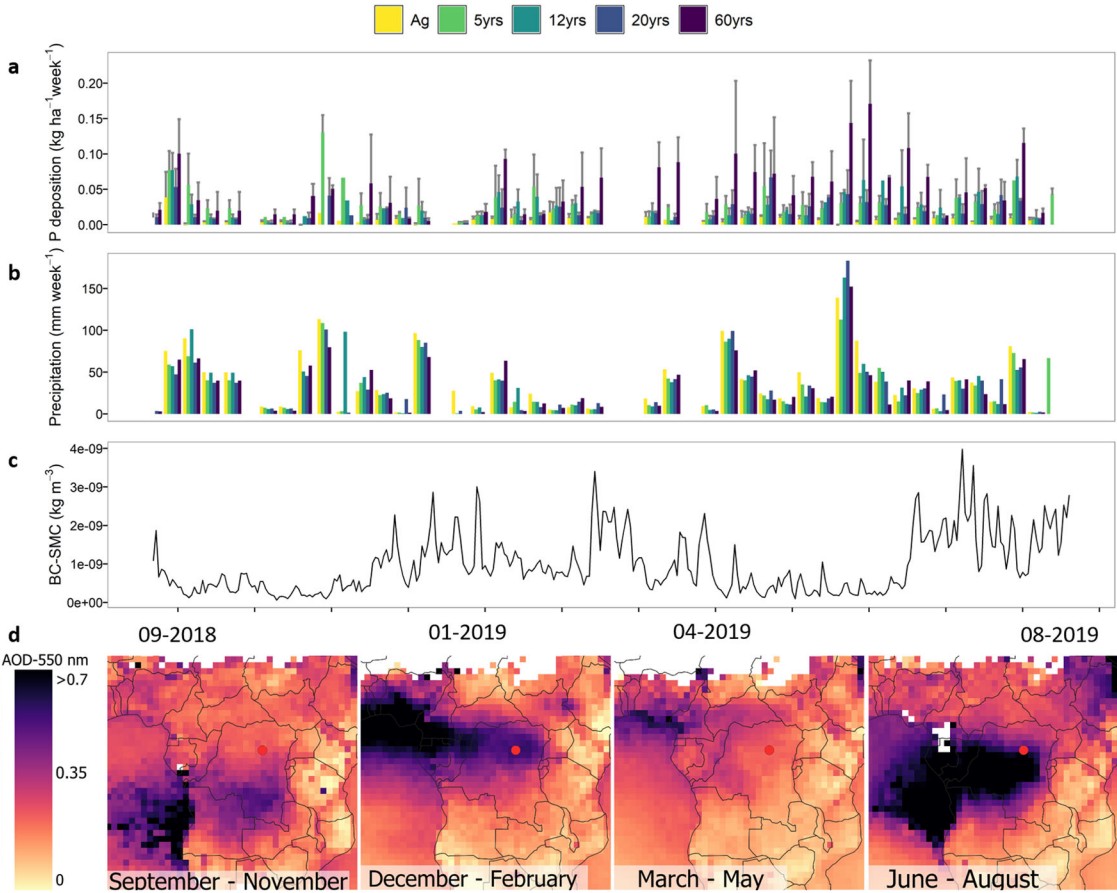

**Fig. 4 Seasonality in phosphorus deposition and black carbon above the monitoring site.** Temporal trends for open field (Ag), 5- (5 yrs), 12- (12 yrs), 20- (20 yrs), and 60-year-old forest (60 yrs) over the monitoring period for phosphorus (P) deposition (mean ± standard deviation) (**a**), weekly precipitation (**b**), black carbon surface mass concentration (BCSMC) (**c**) above our site as extracted from MERRA2. Aerosol optical depth (AOD) at 550 nm, signaling biomass burning-derived aerosols detected over the study period by MODIS' Terra satellite at the larger region around the study site (red circle) (**d**).

equipped with a Hasselblad L1D-20c camera (20 megapixels, 5184 × 3456 pixels, ca. 77° FOV). The onboard GNSS supports GPS and GLONASS. (ii) a customized DJI Phantom 3 Advance. We removed the DJI camera-gimbal system and mounted a GoPro Hero 3 camera (12 megapixels, 4000 × 3000 pixels, with 2.92 mm F/2.8 123° HFOV lens) and connected the camera to a RTK/PPK (real-time kinematic and post-processing kinematic) enabled GNSS receiver to determine camera exposure position at centimeter level. In brief, the Mavic camera provides higher image quality but lower GNSS accuracy, while the GoPro camera provides accurate positioning with PPK solution but had lower resolution of images. The combination of both platforms provides high positional accuracy as well as high-quality images[42]. A 90% forward image overlap and 80% side overlap were used during the flights for both UAV/ camera systems. The flight height was set at 180 m from the ground level, providing an average ground sampling distance (GSD) of ca. 0.04 m px-1 for Mavic and 0.10 m px-1 for GoPro. During the flights, a Reach RS (Emlid Ltd.) base station was mounted on a tripod placed in an open area within the surveyed region to provide correction data for PPK georeferencing. The coordinate of the base station was determined using the ca. 8 h average value of the single solution throughout the survey. The images of the GoPro were georeferenced using the RTKLib© software in PPK mode. This setup provides meter-level absolute accuracy but centimeter-level precision (relative accuracy). The images were then processed using the Pix4D Mapper software (https://www.pix4d.com/) where both image types were combined in a single SfM-workflow. The outputs (i.e., 3D point cloud, DSM and RGB mosaics had a centimetric precision (RMSE estimated at ca. 0.04 m)[42].

**Canopy roughness and canopy rugosity calculation using canopy 3D model**. To show a potential canopy trapping effect, we related the canopy 3D properties to the dry deposition loads in the different plots, using the UAV structure-from-motion 3D product. It has been shown that dry deposition velocity ($v_{ds}$) of particles is linearly related to the surface roughness length ($z_0$)[27,28], and a simple parameterization for various surfaces—across various biomes—has rendered the following relation[27]:

$$v_{ds} = 0.617\log(z_0) + 1.77 \qquad (1)$$

Hence, comparing across the successional stages of our forest succession, the canopy trapping hypothesis should manifest through a consistently different $v_{ds}$ along the chronosequence, if the canopy complexity varies in $z_0$. As such, we aimed to determine $z_0$ along the chronosequence, in order to determine the $v_{ds}$, to see if this related to the differences in dry deposition loads. Hence, $z_0$ was determined following Lettau[43] and De Vries et al.[44]:

$$z_0 = CH\lambda \qquad (2)$$

where $C$ is a constant—assumed 0.5[44], $H$ is the average obstacle height and $\lambda$ is the obstacle density of the roughness elements. $H$ was derived based on field inventories and field-based measurements of tree height. In turn, $\lambda$ was determined following de Vries et al. and Li et al.[44,45] as:

$$\lambda = \frac{\sum \Delta y}{\sum \Delta x} \text{ for } \Delta y > 0 \qquad (3)$$

In other words, $\lambda$ integrates the positive $\Delta y$ divided by the $\Delta x$ over a certain transect of the canopy height model of a certain experimental plot. This was done on 32 transects in the north–south direction of the plots, as well as 32 transects in the east–west direction for every plot. However, the calculation of $\lambda$ over height transects is prone to instrument noise in the canopy height models. To filter out this noise, the transect data can be smoothed by applying a rolling means of a certain size, i.e., including a certain number of neighbor points along the transect. To determine the best size of the rolling mean, the calculated $\lambda$ value can be plotted against the number of measurements that are included in the moving average. The resulting slope of $\lambda$ versus the size of the moving average has two components; a first steep descent, followed by a less steep stable slope. It is accepted that the first steep descent is caused by filtering out unwanted noise in the data, while the second part is the actual $\lambda$ (Fig. S3). Instead of detecting a breakpoint in this curve per plot, we qualitatively inspected the different curves, and selected a general number of measurements to be included in the rolling average (19 points in total, i.e., 9 points on both sides of the point itself) to determine $\lambda$, to avoid that an inconsistent rolling mean would drive differences in the $\lambda$ estimation across plots. The drone flights were performed 6 months after the end of the monitoring period. We do not expect this to affect the forest canopy structure, but the agricultural field were heavily overgrown already by then, compared to the bare fields they were at the beginning of the monitoring period. For this reason we divided the standard deviation of the agricultural plots by three in Fig. 2, for visualization purposes, assuming a linear increase of variability over time, and targeting the average (after 6 months of monitoring) situation in the monitoring period.

Additionally, we calculated canopy rugosity[46]. In essence, this is a much simpler proxy that estimates the standard deviation of the vertical variation of the upper canopy layer. For this, we used transects that were generated based on the CHM of the plots, as described above, to calculate the standard deviation of canopy height. The final 'rugosity' parameter is an average of the standard deviation over all the different transects per plot.

**Data analysis**. We calculated weekly throughfall fluxes of P by multiplying the concentration of P with the weekly recorded rainfall volume per plot. We aggregated this data to annual fluxes by calculating a weekly average and multiplying by 52 weeks per year, to account for some missing values—in case a throughfall collector was stolen, or sampling was logistically impossible that week (on one occasion). Up-to-date, it has proven difficult to quantify net nutrient deposition in forest ecosystems, since most studies are based on throughfall collection. This method uses rainfall collectors set up in open-field, paired with collectors under the forest canopy (i.e., throughfall collection). The measured deposition load in the open field is typically considered 'wet' deposition, while the throughfall collectors are considered to be the sum of wet deposition, dry deposition and canopy leaching, minus canopy uptake of nutrients. Contrary to other direct quantification methods of dry deposition[8,47], this method is attractive for its simplicity and also the potential to fully include canopy complexity effects on the dry deposition budgets. However, the downside of the method is that it is complex to disentangle dry deposition, from a canopy process such as leaching and uptake. There have been a variety of theoretical models that attempt disentangling both exogenous and endogenous effects in throughfall deposition loads, but these show varying success and results (for an overview see ref. [48]). In this study, we approached this differently by combining the deposition loads of a proxy for biomass burning (DBC) with the P deposition loads. Theoretically, subtracting the weekly wet deposition, i.e., the deposition loads as recorded in the open field, from the throughfall deposition load in the different forest types, results in the weekly net canopy effect, i.e., either canopy uptake/leaching or the added dry deposition effect of the canopy. We did this for both DBC deposition loads as well as P deposition loads, and subsequently assessed if the P deposition could be consistently predicted using only DBC deposition loads across the plots. We performed standardized major axis (SMA) regression between DBC deposition load and P deposition load, and between rainfall volume and P deposition. Where assumptions of the model fit were violated, we logtransformed both variables and performed a log–log SMA regression, but these were back-transformed for visualization purposes. SMA regressions were fitted using the 'smatr' package[49], using the robust estimation method. All analyses were carried out with the R software[50].

**Wind trajectories and satellite data**. To show how biomass burning aerosols might be sourced throughout the year across the African continent, we crossed a burnt area dataset with backward wind trajectories ending at the site, similar to previous work at the same site[19]. The fire pixel dataset covering the entire African continent was obtained from NASA's MODIS Collection 6 NRT[51]. The backward wind trajectories were generated using the Hybrid Single Particle Lagrangian Integrated Trajectories (HYSPLIT) model provided by the National Oceanic and Atmospheric Administration Air Resources Laboratory (NOAA ARL)[52–55] with the reanalysis 2.5 degree archive meteorological dataset. For the entire study period, we generated one trajectory at noon for every day, ending above the study site pixel at 500 m above ground level, going back 1 week. Instead of displaying all the trajectories separately, we constructed a convex hull around the trajectories to effectively show the potential source areas to be crossed by winds (Fig. S3). Additionally, we extracted the hourly simulated BCSMC from above our study site from the Modern-era Retrospective Analysis for Research and Applications version 2 (MERRA-2) from the Goddard Earth Observing System Model, version 5[56].

**Uncertainties**. This study reports some of the first in situ measured P deposition loads for central Africa. However, due to logistical constraints and local conditions, we had to design the monitoring setup in a robust and low-tech manner. These adaptations for the long-term monitoring might bring about some additional uncertainties we want to report here. First, the rainfall and throughfall collectors we used in the field did not have cooling (no grid power available, and safety issues in maintaining technological setups in the field), nor did we add preservation chemicals to the collection bottles (due to the environmental risks involved with the disposal of mercury. Nevertheless, we judge that the consequences of this are minor for total phosphorus concentration in the samples. The DBC quantification, by definition, was performed on filtered samples and thus did not include particulate BC (PBC). However, we use DBC merely as a tracer for (dissolved) total P sourced by biomass burning. Given the pre-dominance of biomass burning as the source of combustion-derived aerosols in the Congo (consistent source), the long-range transport of aerosols to our study site, and the sub 0.2 μm size fraction of these aerosols, we are confident that DBC is a reliable tracer for combustion-derived inputs. Finally, we use 'wet' and 'dry' deposition throughout this manuscript, while in fact our wet deposition was not collected with 'wet-only' collectors. In practice, the open-field depositional loads will also have dry deposition collecting on the collector surface. Hence, our 'wet' deposition estimates are slightly overestimating wet deposition, while the 'dry' deposition estimates are slightly underestimating actual dry deposition.

## Data availability
The data generated in this study are provided in the Supplementary information file.

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

## Acknowledgements

We thank our collaborators at the Université de Kisangani and the Université Catholique de Bukavu for the local expertise and support in setting up this research. M.B. is funded by the Fonds Wetenschappelijk Onderzoek (FWO) Flanders, through a post-doctoral fellowship. We thank the people of the Yoko village for the long-lasting collaboration and safeguarding of our equipment in the experimental forest. Some analyses and

visualizations used in this paper were produced with the Giovanni online data system, developed and maintained by the NASA Goddard Earth Sciences Data and Information Services Center. We also acknowledge the MODIS mission scientists and associated NASA personnel for the production of the data used in this research effort. H.V. would like to acknowledge the financial support from the Fonds Wetenschappelijk Onderzoek (FWO; grant no. G018319N). Support was also provided by the National Science Foundation OCE #1756812 to S.W.; T.W.D. was supported by the core funding of ETH Zurich granted to Johan Six. Analyses and visualizations used in this paper were produced with the Giovanni online data system, developed and maintained by the NASA GES DISC. We acknowledge the mission scientists and Principal Investigators who provided the data used in this research effort.

## Author contributions

M.B., T.W.D., and P.B. conceived the study. M.B. and K.V.O. carried out the drone flights. K.V.O. processed the drone imagery for the 3D canopy reconstruction. M.B., C.E., and I.M. oversaw and carried out field work, and set up the permanent sampling units. M.B., T.W.D., S.B., and S.W. analyzed the dissolved black carbon in the composite samples. M.B., I.M., P.B., and S.B. analyzed the phosphorus in the sample set. I.M., C.E., L.C., M. Barthel, and S.B. provided logistical support for the field work. M.B. wrote the paper, with significant contributions from T.W.D., S.W., K.V., H.V., P.B., and K.V.O.

## Competing interests

The authors declare no competing interests.
