## [Peer Review File. · Nature Communications]

Fire-derived phosphorus fertilization of African tropical forestsREVIEWER COMMENTS

Reviewer #1 (Remarks to the Author):

This manuscript presents convincing evidence that canopy complexity of forests of Central Africa plays a key role in the amount of phosphorus stored by these ecosystems. The age of the forest that determines tree structure and the canopy geometry acts as a trap for phosphorus for aged forests. The manuscript is clearly written, the arguments are convincing and I have only some minor points that could be clarified to improve the clarity of the points being made. I recommend that this manuscript be published with these improvements.

Major point: I could not find in the main text or in the Supplement how the ranges for the deposition fluxes were calculated and what uncertainties/variations were accounted for.

Minor points:

Line 81: The measurements you made do not cover whole of Central Africa, refer the reader to Figure S1 that presents the measuring site and shows the extent of the area covered by these measurements.

Line 84: The estimate that you indicate for the region from reference 24 (0.8 to 1.0 kg P ha⁻¹ year⁻¹) did not account for canopy structure. It is interesting to note that this estimate is well within the range of annual deposition of 5-, 12-, and 20-year-old forests ((1.55±0.67 kg P ha⁻¹ yr⁻¹, 1.45±0.28 kg P ha⁻¹ yr⁻¹, and 1.07±0.21 kg P ha⁻¹ yr⁻¹, respectively) that you measure.

Line 111 : You never define UAV, please indicate "Unoccupied Aerial Vehicles".

Line 142: For the BCCMD (black carbon column mass density) from MERRA2, you point the reader to Fig S2. I believe this is a typo and that you meant to point to Fig. 4 of the main text.

Lines 141 to 144: you point to seasonal variations of BCCMD to explain the variations of measured P fluxes. Note that dry deposition is linked to surface concentrations and not to the column load of BC. It would have been more physically correct to show maps of surface concentrations for the dry deposition.

Reviewer #2 (Remarks to the Author):

This manuscript deals with a relevant and challenging issue. The phosphorus dry and wet deposition in tropical forests is not easy to measure and correctly interpret the data. The sampling has significant complications for dry and wet deposition as well as for throughfall. The throughfall is a mixture of wet deposition, canopy leaching as well as dry deposition itself.

More problematic yet is the attribution of fire-associated emissions over forests. We always have long-range transport mixed with local or regional natural biogenic particles. Nutrient recycling is a crucial component in tropical forests, and this is not discussed in the manuscript. The combination of all these factors brings significant uncertainties that require rigorous statistical analysis.

The manuscript has some critical problems. The first one is the lack of a comprehensive statistical treatment of data. The primary data reported in table S1 do not provide experimental errors or standard deviations based on variability. In any modern scientific study, error analysis must be done integrated into the experiment. Tables are presented without standard deviations of the measurements. Plots such as Figure 1, Figure 2, and Figure 3 do not show error bars. Table S2 and S3 also do not include any measurement uncertainties or standard deviations. I think the lack of proper statistical treatment is unacceptable in a scientific study.

The manuscript uses DBC (Dissolved Black Carbon) as a proxy for biomass burning. DBC was quantified using the benzenepolycarboxylic acid (BPCA) method, described in reference 22. This method is a nonstandard method for DBC measurement, and, as this is critically important for this manuscript, it needs to discuss how well his technique provides biomass burning DBC. What about the un-dissolved BC component? The manuscript does not mention this component, and it is essential because BC can have a significant fraction of the mass being non-soluble. It would be essential to measure total BC and compare it with the DBC component. How brown carbon that can account for 20-25% of BC absorption affects their estimate of DBC?.

In the methods section, I think that sample collection has fundamental problems. Rapid degradation of solutions in rainwater samples is very well known, and we usually use Thymol or other biocides to limit algae and bacteria growth. The methods section indicates that Thymol was not used in this study. Also is typical to collect the samples in refrigerated rainwater collectors such as Eigenbrodt wet only rainwater collector (<https://www.eigenbrodt.de/>). This is necessary because if we leave rainwater or throughfall water for a few days in a tropical area, it will decompose very quickly and provide errors in the determination of P and other rainwater components. Just keep the throughfall sampler under the ground do not cool it down enough. Additionally, the study has not used wet only collectors, so the precipitation also included some dry deposition components. This is not even mentioned in the manuscript and needs a much better assessment. Please look at the precipitation protocols of DEBITS (Deposition of biologically important trace species), WMO or EPA, where this critical issue is thoroughly discussed.

The manuscript also calculates canopy roughness using a 3D canopy model to deal with a potential canopy trapping effect. The dry deposition velocity (v_{ds}) of particles over the forest depends on many factors, including vertical air movements and convection. The authors need a better discussion on how these highly uncertain properties affect their results.

The potential biomass burning component needs better discussion about the sources and transport characterization. Most of the fires in Africa are from savannah burning, not fires from pristine tropical deforestation. Of course, the site is affected by BC's long-range transport from savannah fires that are not discussed in the manuscript. A figure with the biomass burning influence regions for the site generated using FLEXPART or HYSPLIT (<https://www.arl.noaa.gov/hysplit/hysplit/>) could add insights on the biomass burning sources for the sampling site.

The manuscript also has some strange descriptions of deforestation versus slash and burning agricultural fields and charcoal production. Two examples in the text that raises confusion and show errors in the drivers of tropical deforestation:

Abstract line 22: Slash and burning agriculture is not a mechanism for the conversion of pristine tropical forests. Deforestation is the mechanism involving pristine forests. *Slash-and-burn agriculture will convert important areas of pristine forest to fallow-based agroecosystems and secondary forest.*

Line 58 - Curiously, the manuscript does not recognize deforestation as a mechanism affecting tropical forests. Only slash and burn agriculture and charcoal production. Congo is the third country in terms of deforestation, after Brazil and Indonesia. *“The African continent represents nearly 65% of the global burnt area annually¹⁷, in part due to the prevalence of slash-and-burn agriculture^{18,19} and associated high rates of charcoal production²⁰ “*. The 65% of the global burnt area is due mainly to savannah fires, not tropical forest fires. I think this must be corrected.

In the abstract and the final discussions, the authors extrapolate this single point measurement to the Congo Tropical forests. Of course, this should not be done since the forest is quite heterogeneous, and one single point of measurement in a million km² is not representative of the whole forest.

Based on the above discussion, I think the manuscript has serious issues that maybe could be corrected in a revised version. At least it requires a profound revision on methods, data analysis, and conclusions. As it stands now, I do not recommend this manuscript for publication in Nature Comm.

Reviewer #3 (Remarks to the Author):

This is a very interesting study of P input in tropical forests in regions with disturbances. They show that slash-and-burn practices export P (and probably other base cation nutrients) as fire derived dry

deposition to the surrounding forests in amounts that may lift P limitation. The study is thorough including 12 forest plots and 3 agricultural fields in tropical Africa (Congo) – an understudied region. It will make a valuable contribution to the understanding of tropical forests and the anthropogenic impact on these ecosystems.

The paper is well written and the methods describe in sufficient detail. However I have a few major issues to be addressed in a revision:

1) The effect of canopy roughness (Fig 2) which has a central role in the paper is not convincing since I do not think there is a relationship between P-input and the canopy roughness parameter when the agricultural fields are removed. Could some of the effect be masked by variable distances to burned source areas? The observations can as they appear now be taken as an indication not as proof the complexity increase P deposition.

Just some thoughts on the canopy roughness parameter. It is not my expertise but recently fell over the parameter 'rugosity' (see e.g. doi:10.1016/j.foreco.2003.09.001 and an example use of it in Hardiman et al 2013 <https://doi.org/10.1016/j.foreco.2013.02.031>), just wonder how it compares to what was done here and if alternative approaches have been considered.

2) Line 119-126: This is a kind of circular argument: that slash-and-burn remove canopy which then reduce P-input that is released by slash-and-burn. I am not sure this is an important issue.

Reformulate....

3) Although the methods describe the deposition parameters in detail (throughfall, wet, dry etc.) the text and Fig. 1 could be more clear in defining what is discussed, particularly dry deposition which appear to be THF minus BP.

4) Some minor comments and removal of meaningless decimals on fluxes that vary can be found in the attached file.

Dear Dr Bauters,

First, please let me apologize again for the time it has taken to get back to you with a decision.

Thank you again for submitting your manuscript "Fire-derived phosphorus fertilization of African tropical forests" to Nature Communications. We have now received reports from 3 reviewers and, after careful consideration, we have decided to invite a major revision of the manuscript.

As you will see from the reports copied below, the reviewers raise important concerns. We find that these concerns limit the strength of the study, and therefore we ask you to address them with additional work. Without substantial revisions, we will be unlikely to send the paper back to review. In particular, Reviewer #2 raised a number of critical concerns regarding the methods, which will need to be justified and discussed more thoroughly. Further, this reviewer also stressed the need for error and uncertainty reporting. Reviewer #3 also raised some concerns about the relationship between P-input and canopy roughness that needs to be addressed.

We thank the editor for their deft handling of our manuscript and appreciate the challenges of finding reviewers these days. We are completely confident in our ability to address the concerns of the reviewers. Please find our detailed responses below. To facilitate the review process, all line numbers refer to changes in the track-changed version of the manuscript, attached at the end of this rebuttal letter.

REVIEWER COMMENTS

Reviewer #1 (Remarks to the Author):

This manuscript presents convincing evidence that canopy complexity of forests of Central Africa plays a key role in the amount of phosphorus stored by these ecosystems. The age of the forest that determines tree structure and the canopy geometry acts as a trap for phosphorus for aged forests. The manuscript is clearly written, the arguments are convincing and I have only some minor points that could be clarified to improve the clarity of the points being made. I recommend that this manuscript be published with these improvements.

We thank the reviewer for the constructive and positive assessment of our manuscript. We have now used the suggestions to create an improved MS version.

Major point: I could not find in the main text or in the Supplement how the ranges for the deposition fluxes were calculated and what uncertainties/variations were accounted for.

This is now explained in the first sentences of the section "Data analysis" and we have added a short sentence in the main text to make this clear to the readers. Additionally, we added a new 'uncertainties' paragraph to the end of the methods section to make clear to the reader what the potential uncertainties of these setups might be. (L 69-76, L365-382)

Minor points:

Line 81: The measurements you made do not cover whole of Central Africa, refer the reader to Figure S1 that presents the measuring site and shows the extent of the area covered by these measurements.

The reviewer is right, we now explicitly added this info in the reporting. (L94)

Line 84: The estimate that you indicate for the region from reference 24 (0.8 to 1.0 kg P ha⁻¹ year⁻¹) did not account for canopy structure. It is interesting to note that this estimate is well within the range of annual deposition of 5-, 12-, and 20-year-old forests ((1.55±0.67 kg P ha⁻¹ yr⁻¹, 1.45±0.28 kg P ha⁻¹ yr⁻¹, and 1.07±0.21 kg P ha⁻¹ yr⁻¹, respectively) that you measure.

Indeed, very interesting, especially given that these numbers were derived from modeling work! We now also stress this in the manuscript (L123-126): “Annual P deposition in the 60-year-old forest (3.1±1.4 kg P ha⁻¹ yr⁻¹; Fig. 2) was found to be twice that of annual P deposition in 5-, 12-, and 20-year-old forests, which exhibited similar values (1.6±0.7 kg P ha⁻¹ yr⁻¹, 1.5±0.3 kg P ha⁻¹ yr⁻¹, and 1.1±0.2 kg P ha⁻¹ yr⁻¹, respectively) which in turn relate well to simulated estimates of annual P deposition for the area²⁴.”

Line 111 : You never define UAV, please indicate “Unoccupied Aerial Vehicles”.

We thank the reviewer for this suggestion, we have now added this to the new MS version. (L128)

Line 142: For the BCCMD (black carbon column mass density) from MERRA2, you point the reader to Fig S2. I believe this is a typo and that you meant to point to Fig. 4 of the main text.

Indeed, the reviewer is right, this was an error. We now corrected this in the new MS version. (L163)

Lines 141 to 144: you point to seasonal variations of BCCMD to explain the variations of measured P fluxes. Note that dry deposition is linked to surface concentrations and not to the column load of BC. It would have been more physically correct to show maps of surface concentrations for the dry deposition.

Excellent remark. The reviewer is right, we have now redone the analysis with the black carbon surface mass density, and made changes accordingly throughout the manuscript. As you see, this does not affect the story – the trend is highly similar. (New Figure 4, L163,166, 169)

Reviewer #2 (Remarks to the Author):

This manuscript deals with a relevant and challenging issue. The phosphorus dry and wet deposition in tropical forests is not easy to measure and correctly interpret the data. The sampling has significant complications for dry and wet deposition as well as for throughfall. The throughfall is a mixture of wet deposition, canopy leaching as well as dry deposition itself. More problematic yet is the attribution of fire-associated emissions over forests. We always have long range transport mixed with local or regional natural biogenic particles. Nutrient recycling is a crucial component in tropical forests, and this is not discussed in the manuscript. The combination of all these factors brings significant uncertainties that require rigorous statistical analysis.

We thank the reviewer for the constructive and thorough assessment of the manuscript. The reviewer has good suggestions, but we also feel that we can defend our approach in other points. Below we respond on a point-by-point basis to the reviewer. We have changed the manuscript where we agreed with the reviewer and feel that this has greatly improved the manuscript quality.

The manuscript has some critical problems. The first one is the lack of a comprehensive statistical treatment of data. The primary data reported in table S1 do not provide experimental errors or standard deviations based on variability. In any modern scientific study, error analysis must be done integrated into the experiment. Tables are presented without standard deviations of the measurements. Plots such as Figure 1, Figure 2, and Figure 3 do not show error bars. Table S2 and S3 also do not include any measurement uncertainties or standard deviations. I think the lack of proper statistical treatment is unacceptable in a scientific study.

We are confused by this comment.

Supplementary Tables: Table S1; this is all plot-level data. The deposition data is accumulated deposition data (originating from 52 summed weekly samples). The true replicates are the triplicated plots per stage. Instead of summarizing the data per successional stage and showing standard deviations per successional stage, we feel it is much more informative to report the raw plot-level data. The same is true for Table S2 and S3: these are the raw data that underlie the figures and regression models we fitted for the paper: i.e. the individual sample values. We do not feel it is appropriate to summarize these data to plot level and give averages +/- standard deviations given Nature Publishing's policy to provide the raw data in supplementary files as much as possible.

Main Figures: For Figure 1, 2, and 3 we are left with the same confusion: these depict the regression models on individual point measurements. Rather than only showing boxplots (which are added on the side panels of the regression figure, including percentile whiskers), we show the raw data to leave the reader with a good sense of the spread. We are thus unclear as to what kind of error bars the reviewer suggests we should include for the points, since these are individual measurements. Maybe the confusion stems from Figure 1, where only values are given for the general successional stages ('5yrs', '12 yrs' etc.)? As explained in the methods section, the samples of the three replicated plots per stage were composited for DBC analysis, so again, these points represent individual measurements of composite samples.

To avoid that potential source of confusion, we have now explicitly stated within each figure caption what the individual points represent. For Figure 4, we have added error bars for the deposition data, since here this is shown on the successional stage-level. Additionally, R^2 values are reported specifically in the figures to give the readers an informed 'first-look' of what the goodness-of-fit is for the data, for Figure 1 and Figure 2. For Figure 3 we have now added a table (Table S4) with the respective R^2 values and model equations in supplementary, because the figure itself would become too busy.

The manuscript uses DBC (Dissolved Black Carbon) as a proxy for biomass burning. DBC was quantified using the benzenepolycarboxylic acid (BPCA) method, described in reference 22. This method is a nonstandard method for DBC measurement, and, as this is critically important for this manuscript, it needs to discuss how well his technique provides biomass burning DBC.

We thank the reviewer for her/his important questions regarding DBC. We have added the following text to emphasize that the BPCA method quantifies the condensed aromatic fraction of DBC and is the leading technique for quantifying DBC in aquatic systems, including throughfall and stemflow (L245-254):

"The BPCA approach to quantifying DBC involves chemothermal oxidation of condensed aromatic DOC compounds to benzenhexacarboxylic acid (B6CA) and benzenepentacarboxylic acid (B5CA) products. The B6CA and B5CA oxidation products are robustly measured and derive exclusively from pyrogenic sources (Kappenberg et al., 2016). Condensed aromatic DBC, as measured using the BPCA method, is ubiquitous in aquatic environments globally (Jones et al., 2020; Coppola and Druffel, 2016; Wagner et al., 2018; 2019a). DBC has also been quantified in throughfall and stemflow in longleaf pine forests that undergo regular prescribed burning (Wagner et al., 2019b). Therefore, we use the BPCA method as a proxy for carbon inputs from biomass burning in the current study."

References (also added to the manuscript now)

Jones, M.W., Coppola, A.I., Santin, C., Dittmar, T., Jaffe, R., Doerr, S.H., Quine, T.A. Fires prime terrestrial organic carbon for riverine export to the global oceans. *Nat. Commun.* 11, 2791, doi.org/10.1038/s41467-020-16576-z (2020).

Kappenberg, A., Bläsing, M., Lehdorff, E., & Amelung, W. (2016). Black carbon assessment using benzene polycarboxylic acids: limitations for organic-rich matrices. *Organic Geochemistry*, 94, 47–51. <https://doi.org/10.1016/j.orggeochem.2016.01.009>

Coppola, A.I., Druffel, E.R.M. Cycling of black carbon in the ocean. *Geophys. Res. Lett.* 43, 4477–4482, doi:10.1002/2016GL068574 (2016).

Wagner, S., Brandes, J., Spencer, R.G.M., Ma, K., Rosengard, S.Z., Moura, J.M.S., Stubbins, A. Isotopic composition of oceanic dissolved black carbon reveals non-riverine source. *Nat. Commun.* 10, 5064, doi.org/10.1038/s41467-019-13111-7 (2019).

Wagner, S., Jaffé, R., & Stubbins, A. (2018). Dissolved black carbon in aquatic ecosystems. *Limnology and Oceanography Letters*, 3(3), 168–185. <https://doi.org/10.1002/lol2.10076>

Wagner, S., Brantley, S., Stuber, S., Van Stan, S., Whitetree, A., Stubbins, A. (2019) Dissolved black carbon in throughfall and stemflow in a fire-managed longleaf pine woodland. *Biogeochemistry* 146:191–207. doi.org/10.1007/s10533-019-00620-2

What about the un-dissolved BC component? The manuscript does not mention this component, and it is essential because BC can have a significant fraction of the mass being non-soluble. I would be essential to measure total BC and compare it with the DBC component. How brown carbon that can account for 20-25% of BC absorption affects their estimate of DBC?

We agree that undissolved, or particulate black carbon (PBC) is a potentially interesting component of deposition. However, we were solely interested in using DBC as a proxy for fire-derived depositional inputs rather than quantifying the total inputs of BC. Furthermore, the mass median diameter (MMD) of BC aerosols is ubiquitously at or below 0.2 micron (see rebuttal table below for some literature values). This size is well below the threshold for DBC in our samples that were filtered to 0.45 micron and thus supports our use of DBC as a proxy for quantity of fire-derived inputs. Lastly, given the predominance of biomass burning as the source of combustion derived aerosols in the Congo and the long-range transport of aerosols to our study site, there is no reason to expect the proportions of DBC and PBC to vary substantially through time. We have added this also to the ‘uncertainties’ paragraph at the end of the methods section. (L365-382)

Regarding the brown and black carbon analytical overlap: naming conventions used to describe thermally altered carbon fractions have evolved alongside the methods used to characterize them (Hammes and Abiven, 2013). For example, researchers in the atmospheric community describe BC as light-absorbing, soot-like aerosols, whereas aquatic scientists typically describe BC as the condensed aromatic carbon fraction measured via BPCA analysis (see reply to previous comment). We now describe the analytical windows for the method used to quantify DBC in the current study to clarify exactly what we are measuring here. However, it is unknown how DBC as measured by BPCAs overlaps with brown carbon measurements the reviewer refers to here.

References

Hammes, K. & Abiven, S. Identification of Black Carbon in the Earth System. in *Fire Phenomena and the Earth System: An Interdisciplinary Guide to Fire Science* (2013). doi:10.1002/9781118529539.ch9

Particle Source	BC MMD (nm)	Reference
BB (long range transport)	120-160	Dahlkötter et al. 2014
BB (Asia)	177-197	Kondo et al. 2011
BB (Canada)	176-238	Kondo et al. 2011
BB (California)	177-209	Sahu et al. 2012
BB (Texas)	200-220	Schwarz et al. 2008
BB (Canada)	195	Taylor et al. 2014
BB (Wyoming)	170	Pratt et al. 2011
Urban Emissions (California)	120-160	Metcalf et al. 2012
Urban Emissions (Texas)	160-180	Schwarz et al. 2008

Rebuttal Table 1: reported mass median diameter (MMD) of black carbon (BC). Adopted from Williams et al. 2019.

Table references:

Dahlkötter, F., Gysel, M., Sauer, D., Minikin, A., Baumann, R., Seifert, P., et al. (2014). The Pagami Creek smoke plume after long-range transport to the upper troposphere over Europe – Aerosol properties and black carbon mixing state. *Atmospheric Chemistry and Physics*, 14(12). <https://doi.org/10.5194/acp-14-6111-2014>

Kondo, Y., Matsui, H., Moteki, N., Sahu, L., Takegawa, N., Kajino, M., et al. (2011). Emissions of black carbon, organic, and inorganic aerosols from biomass burning in North America and Asia in 2008. *Journal of Geophysical Research Atmospheres*, 116(8), 1–25. <https://doi.org/10.1029/2010JD015152>

Sahu, L. K., Kondo, Y., Moteki, N., Takegawa, N., Zhao, Y., Cubison, M. J., et al. (2012). Emission characteristics of black carbon in anthropogenic and biomass burning plumes over California during ARCTAS-CARB 2008. *Journal of Geophysical Research Atmospheres*, 117(16), 1–20. <https://doi.org/10.1029/2011JD017401>

Schwarz, Joshua P., Gao, R. S., Spackman, J. R., Watts, L. A., Thomson, D. S., Fahey, D. W., et al. (2008). Measurement of the mixing state, mass, and optical size of individual black carbon particles in urban and biomass burning emissions. *Geophysical Research Letters*, 35(13), 1–5. <https://doi.org/10.1029/2008GL033968>

Taylor, J. W., Allan, J. D., Allen, G., Coe, H., Williams, P. I., Flynn, M. J., et al. (2014). Sizedependent wet removal of black carbon in Canadian biomass burning plumes. *Atmospheric Chemistry and Physics*, 14(24), 13755–13771. <https://doi.org/10.5194/acp-14-13755-2014>

Pratt, K. A., Murphy, S. M., Subramanian, R., Demott, P. J., Kok, G. L., Campos, T., et al. (2011). Flight-based chemical characterization of biomass burning aerosols within two prescribed burn smoke plumes. *Atmospheric Chemistry and Physics*, 11(24), 12549–12565. <https://doi.org/10.5194/acp-11-12549-2011>

Metcalf, A. R., Craven, J. S., Ensberg, J. J., Brioude, J., Angevine, W., Sorooshian, A., et al. (2012). Black carbon aerosol over the Los Angeles Basin during CalNex. *Journal of Geophysical Research Atmospheres*, 117(8), 1–24. <https://doi.org/10.1029/2011JD017255>

Williams, Walt, "Airborne Characterization of Black Carbon Aerosol in California from Biomass Burning" (2019). All Theses. 3164. https://tigerprints.clemson.edu/all_theses/3164

In the methods section, I think that sample collection has fundamental problems. Rapid degradation of solutions in rainwater samples is very well known, and we usually use Thymol or other biocides to limit algae and bacteria growth. The methods section indicates that Thymol was not used in this study. Also is typical to collect the samples in refrigerated rainwater collectors such as Eigenbrodt wet only rainwater collector (<https://www.eigenbrodt.de/>). This is necessary because if we leave rainwater or throughfall water for a few days in a tropical area, it will decompose very quickly and provide errors in the determination of P and other rainwater components. Just keep the throughfall sampler under the ground do not cool it down enough. Additionally, the study has not used wet only collectors, so the precipitation also included some dry deposition components. This is not even mentioned in the manuscript and needs a much better assessment. Please look at the precipitation protocols of DEBITS (Deposition of biologically important trace species), WMO or EPA, where this critical issue is thoroughly discussed.

We agree with the reviewer that using thymol is generally a good practice for these kinds of setups. However, in this case, we designed the project specifically without adding thymol, because we are basing all our findings on total phosphorus (TP) concentration of the sample. The reviewer is right that the elemental species composition of the sample might change through microbial activity: i.e. relative shift in N species composition, DOC decomposition etc. However, total phosphorus concentrations of the samples are not likely to be altered by microbial activity: the total P amount in the bottle will not change through microbial activity. Additionally, we explicitly wanted to avoid adding unnecessary organic chemicals to the samples. Thymol contains an aromatic ring, which might contaminate our future planned FT-ICR MS analyses for the samples (Bauters et al. 2018). We considered using mercury, but the risk for environmental contamination (the labs in the DRC are not equipped for the disposal) was simply too high. Putting the collectors in the ground and covering them was not only done to cool them down as much as possible, but also to avoid sun light (algal growth).

For the automated collectors, please bear in mind that these field sites are remote and there is no grid power in the forest. Additionally, the more the setup relies on advanced and expensive technologies (incl. solar panels, powered collectors, etc.), the more problems we have with maintaining and protecting it from theft and intermittent failures. We have operated one ISCO automated sampler (for peak discharge) at this site for 1 year now and it has been stolen once (but we recovered it) and the batteries have been stolen twice. You can imagine the difficulties of trying to maintain 120 collectors (15 x 8) with similar theft rates. Hence, we have learnt the hard way that low-tech approaches are preferred to ensure continued on-site monitoring.

*Regardless, we now acknowledge specifically in the manuscript that there might have been some microbial decomposition because of our 1-week sample storage in the field, but state explicitly also that this we do not expect this to affect the **total phosphorus concentrations** of the samples. We have now explicitly acknowledged this, along with the wet vs. dry only collectors, in the new 'uncertainties' section at the end of the methods section. (L365-382)*

The manuscript also calculates canopy roughness using a 3D canopy model to deal with a potential canopy trapping effect. The dry deposition velocity (v_{ds}) of particles over the forest depends on many factors, including vertical air movements and convection. The authors need a better discussion on how these highly uncertain properties affect their results. The potential biomass burning

component needs better discussion about the sources and transport characterization. Most of the fires in Africa are from savannah burning, not fires from pristine tropical deforestation. Of course, the site is affected by BC's long-range transport from savannah fires that are not discussed in the manuscript. A figure with the biomass burning influence regions for the site generated using FLEXPART or HYSPLIT (<https://www.arl.noaa.gov/hysplit/hysplit/>) could add insights on the biomass burning sources for the sampling site.

For the dry deposition velocity: Indeed, high resolution wind speed and direction data would be needed to simulate the actual (high resolution) dry deposition. However, given that our study is looking deposition at low temporal resolution (accumulated, weekly), and given that the proximity of the plots controls for any significantly divergent wind conditions over the landscape, we feel that the structural component (canopy roughness/rugosity) can be isolated as the dominant driver for the dry deposition. This is also why we focus on this in the manuscript and why the statistical fits perform so well. Given these factors, we do not find it necessary to delve into actual high resolution dry deposition velocity for this study.

Considering the dry deposition velocity: we now added a short section on this in the new 'uncertainty paragraph'. For the source of the biomass burning-derived deposition: we agree with the reviewer that this was not discussed explicitly enough in the previous version. In fact, we have exploited the combination of HYSPLIT and fire incidence in Africa in previous work on the same site (but only for pristine forest and a different monitoring period; Bauters et al. 2018). While this is indeed informative, we don't feel that we can do a better job with this dataset than in the previous work on identifying sources, since it would basically just repeat the same analysis for a different paper. Additionally, through substantial storage of aerosol constituents in the canopy, as we show, the signal becomes obscured over the year. However, we now more explicitly refer to the findings of this earlier work and have added a new Figure S3 in supplementary material along with a new methods section, in which we apply the same methodology (HYSPLIT vs. Fire incidence per season) to show to the reader how biomass burning aerosols could be sourced and transported across the continent. (L351-363, new Figure S3, L108)

References:

Bauters, M. et al. High fire-derived nitrogen deposition on central African forests. Proc. Natl. Acad. Sci. 115, 549–554 (2018).

The manuscript also has some strange descriptions of deforestation versus slash and burning agricultural fields and charcoal production. Two examples in the text that raises confusion and show errors in the drivers of tropical deforestation: Abstract line 22: Slash and burning agriculture is not a mechanism for the conversion of pristine tropical forests. Deforestation is the mechanism involving pristine forests. Slash-and-burn agriculture will convert important areas of pristine forest to fallow-based agroecosystems and secondary forest. Line 58 - Curiously, the manuscript does not recognize deforestation as a mechanism affecting tropical forests. Only slash and burn agriculture and charcoal production. Congo is the third country in terms of deforestation, after Brazil and Indonesia.

We thank the reviewer for pointing out this unclear use of terminology. In our original manuscript, we were mainly focusing on slash-and-burn and charcoal production, since these are the main drivers for deforestation in the Congo basin, as opposed to South America and South-east Asia, where commercial logging and commodity plantations are a far bigger threat for pristine forests (see Curtis et al. 2018). In central Africa, deforestation, i.e. conversion of pristine forest to other land use types, is predominantly caused by smallholder practices such as slash and burn, and charcoal production, as has been shown by several studies (Curtis et al. 2018, Tyukavina et al. 2018, Tyukavina et al. 2013).

We have now rephrased our statements in several parts of the manuscript to avoid confusion.(L21-28, L60-62)

References:

*Curtis, P. G., Slay, C. M., Harris, N. L., Tyukavina, A. & Hansen, M. C. Classifying drivers of global forest loss. Science (80-.). **1111**, 1108–1111 (2018).*

*Tyukavina, A. et al. Congo Basin forest loss dominated by increasing smallholder clearing. Sci. Adv. **4**, (2018).*

*Tyukavina, A. et al. National-scale estimation of gross forest aboveground carbon loss: a case study of the Democratic Republic of the Congo. Environ. Res. Lett. **8**, 044039 (2013).*

“The African continent represents nearly 65% of the global burnt area annually¹⁷, in part due to the prevalence of slash-and-burn agriculture^{18,19} and associated high rates of charcoal production²⁰ “. The 65% of the global burnt area is due mainly to savannah fires, not tropical forest fires. I think this must be corrected.

We fully agree with the reviewer on this point. We have now changed this in the new MS version. (L61-62)

In the abstract and the final discussions, the authors extrapolate this single point measurement to the Congo Tropical forests. Of course, this should not be done since the forest is quite heterogeneous, and one single point of measurement in a million km² is not representative of the whole forest. Based on the above discussion, I think the manuscript has serious issues that maybe could be corrected in a revised version. At least it requires a profound revision on methods, data analysis, and conclusions. As it stands now, I do not recommend this manuscript for publication in Nature Comm.

We agree with the reviewer that the previous manuscript version might have overstated the results in some parts. We have now changed this in the new MS version and better qualified our extrapolation (L94, L30, L37, L38). Nevertheless, we want to stress that the main finding of manuscript is the mechanism by which more complex and mature forests capture dry deposition. Although the reviewer states that the Congo forests are quite heterogeneous, recent research has shown that the forests of the whole Congo Basin can be classified in three rather homogenous types (Réjou-Méchain et al. 2021 Nature). Additionally, forest heterogeneity per se (in terms of species distributions etc.) should not matter for P deposition. Rather, it is the combination of forest structure and the cocktail of biomass burning aerosols and its spatial (and temporal) fluctuations that will ultimately determine the magnitude of P deposition. The forest structure link is clearly exploited in our manuscript (in a mechanistic way), and wanted to show via Figure 4d that biomass burning aerosols are omnipresent in large parts of the basin. Hence, although we indeed tone down the overgeneralization in the new MS version, we feel confident that the mechanisms that underpin our observations hold for a much larger region.

Reference:

Réjou-Méchain, M. et al. Unveiling African rainforest composition and vulnerability to global change. Nature (2021). doi:10.1038/s41586-021-03483-6

In our responses and revised manuscript, we feel we were able to further clarify our approach in several parts of the paper's methodology and amend our analyses and conclusions.

Reviewer #3 (Remarks to the Author):

This is a very interesting study of P input in tropical forests in regions with disturbances. They show that slash-and-burn practices export P (and probably other base cation nutrients) as fire derived dry deposition to the surrounding forests in amounts that may lift P limitation. The study is thorough including 12 forest plots and 3 agricultural fields in tropical Africa (Congo) – an understudied region. It will make a valuable contribution to the understanding of tropical forests and the anthropogenic impact on these ecosystems. The paper is well written and the methods describe in sufficient detail.

We thank the reviewer for the constructive and positive assessment of our manuscript. We have now used the suggestions to create an improved MS version.

However I have a few major issues to be addressed in a revision:

1) The effect of canopy roughness (Fig 2) which has a central role in the paper is not convincing since I do not think there is a relationship between P-input and the canopy roughness parameter when the agricultural fields are removed. Could some of the effect be masked by variable distances to burned source areas? The observations can as they appear now be taken as an indication not as proof the complexity increase P deposition.

Just some thoughts on the canopy roughness parameter. It is not my expertise but recently fell over the parameter 'rugosity' (see e.g. doi:10.1016/j.foreco.2003.09.001 and an example use of it in Hardiman et al 2013 <https://doi.org/10.1016/j.foreco.2013.02.031>), just wonder how it compares to what was done here and if alternative approaches have been considered.

We thank the reviewer for this excellent suggestion. In fact, yes, we considered including 'rugosity' in the original manuscript version. The paper referred to (Hardiman et al.) used a lidar (i.e. vertical canopy profile) based proxy, that integrates the full 3D 'heterogeneity' of the canopy structure. For our sites, these LiDAR measurements are not available, nor does it seem necessary when considering physics behind dry deposition. This is because the lower canopy layers should play a relatively minor role for the deposition of particles on the canopy surface layer. However, Hardiman et al. derived their method from Parker and Russ (reference below), who only considered the upper canopy surface 'vertical height variability'. This is an excellent metric to compare to our 'roughness' parameter. In some ways, it is a simplified version of our 'roughness index: it is just a proxy for the overall variability in height along the canopy transects (cf. Figure S2 in the original manuscript file). Originally, we felt that our roughness parameter was a better 'mechanistic' parameter to include, because it is a proxy for the likelihood that a particle encounters a positive height difference (an obstacle) as it passes over the canopy. However, it seems like rugosity – a much simpler index – might also be worth including for simply being much easier to grasp. Hence, although we decided not to include this in the original manuscript version, we feel that many readers might be left with the same question of the reviewer, so we now also include this in the revised version. (Changed Figure 2, L131-133, L320-324)

Regarding the agricultural fields potentially driving the relationship: actually, the relationship holds without the fields (see Rebuttal figure 1 below). Nevertheless, we generally downturned our conclusions and reformulated these relations (although rugosity again confirms a good fit in our opinion) as being an indication for rather than hard proof. (L30, L32, L131-137, L182)

References

Parker, G. G. & Russ, M. E. The canopy surface and stand development : assessing forest canopy structure and complexity with near-surface altimetry. *For. Ecol. Manage.* **189**, 307–315 (2004).

Rebuttal Figure 1: Canopy Roughness as a predictor for P dry deposition. This figure shows the linear fits with (upper panel) and without (lower panel) agricultural fields. Note that the effect estimate hardly changes with omission of the agricultural fields.

2) Line 119-126: This is a kind of circular argument: that slash-and-burn remove canopy which then reduce P-input that is released by slash-and-burn. I am not sure this is an important issue. Reformulate....

We thank the reviewer for pointing this out. We have now indeed reformulated this in the new MS version. This was too speculative for our own data, but the statement actually just tries to provide the reader with an 'outlook' of our observations, based on similar work by Lawrence and colleagues. We believe this is now better formulated in the new version. (L142-152)

3) Although the methods describe the deposition parameters in detail (throughfall, wet, dry etc.) the text and Fig. 1 could be more clear in defining what is discussed, particularly dry deposition which appear to be THF minus BP.

We thank the reviewer for pointing this out. We now more clearly state in the main text as well as in the figure caption what we considered 'dry deposition' and how this was calculated and added explicit pointers on several places to remind the reader of this. We also acknowledge in the new 'uncertainties' section that our dry and wet deposition quantification are not exact because we did not use 'wet only' collectors. (L69-76, L103-104, L154-155)

4) Some minor comments and removal of meaningless decimals on fluxes that vary can be found in the attached file.

We thank the reviewer for these comments. This will improve the readability and consistency of the manuscript. We have made following edits as suggested:

- *Removed decimals L96*
- *Added 'dry' L101*
- *Removed decimals L123, 124, 125*
- *Deleted space L190*
- *Removed decimals L215*
- *Removed decimals L264*
- *Deleted space L2*

**Title: Fire-derived phosphorus fertilization of African tropical forests**

**Authors:** Marijn Bauters^{1,2,*}, Travis W. Drake³, Sasha Wagner⁴, Simon Baumgartner^{1,5}, Isaac
Makelele^{1,6}, Samuel Bodé¹, Kris Verheyen², Hans Verbeeck², Corneille Ewango⁷, Landry
Cizungu⁸, Kristof Van Oost⁵, Pascal Boeckx¹

¹ Department of Green Chemistry and Technology, Ghent University, Ghent, 9000, Belgium

² Department of Environment, Ghent University, Ghent, 9000, Belgium

³ Department of Environmental Systems Science, Swiss Federal Institute of Technology, ETH
Zurich, Zurich, 8092, Switzerland

⁴ Department of Earth and Environmental Sciences, Rensselaer Polytechnic Institute, Troy,
New York, USA

⁵ Earth and Life Institute, UCLouvain, Louvain-la-Neuve, 1348, Belgium

⁶ Department of Biology, Université Officielle de Bukavu, Bukavu, DR Congo

⁷ Faculty of Renewable Natural Resources Management, University of Kisangani, B.P.O.
2012, Kisangani, DR Congo

⁸ Soil science laboratory, Faculty of Agronomy, Université Catholique de Bukavu, Bukavu,
DR Congo

* Corresponding author: Marijn.Bauters@UGent.be

~~The Congo basin's~~ Central African tropical forests ~~is face-facing~~ increasing
anthropogenic pressures, particularly in the form of deforestation and land-use
conversion to agriculture. The long-term effects of this transformation of ~~-, converting~~
~~substantial areas of pristine forests - to~~ fallow-based agroecosystems and secondary
forests ~~secondary forests through biomass burning-associated practices. Slash-and-burn~~
~~agriculture will convert substantial areas of pristine forest to fallow-based~~
~~agroecosystems and secondary forest.~~ The long-term effect of such anthropogenic
~~activities~~ on biogeochemical cycles that drive forest functioning are poorly understood.
Here, we show that biomass burning ~~activities~~ on the African continent results in high
phosphorus (P) deposition on ~~equatorial forest~~ an equatorial forests via fire-derived
atmospheric emissions. Furthermore, we show that deposition loads increase with forest
regrowth age, likely due to increasing canopy complexity, ranging from 0.4 kg P ha⁻¹ yr⁻¹
on agricultural fields to 3.1 kg P ha⁻¹ yr⁻¹ on old secondary forests. In forest systems,
canopy wash-off of dry P deposition increases with rainfall amount, highlighting how
tropical forest canopies act as dynamic reservoirs for enhanced addition of this essential
plant nutrient. Overall, the magnitude of the observed P deposition load at the study site
challenges the P-limitation status ~~of the second-largest tropical forest of this forest on~~
~~Earth~~ and demonstrates the importance of canopy trapping as a pathway for nutrient
input into forest ecosystems.

Tropical forests store a substantial amount of the Earth's terrestrial carbon and host a large
share of global biodiversity. Although nitrogen (N) is a potentially limiting plant nutrient
early-on in tropical forest succession^{1,2}, it is a virtually infinite resource that can be fixed
biologically from air³⁻⁵. In contrast, phosphorus (P) has been identified to be the main limiting
nutrient in many old-growth tropical forests rooted in strongly weathered soils^{6,7}. As such,

these forest ecosystems are dependent upon trace inputs of P and cations from the atmosphere
(e.g., as dust or biomass-derived emissions) rather than inputs from soil weathering^{8,9}. In light
of the dependency of tropical forests on these atmospheric P inputs, it is hypothesized that
forest canopies may function as a ‘trap’ for atmospheric constituents^{10,11}. As they regrow,
secondary forests exhibit increased canopy complexity (e.g., surface area or roughness),
which enables more efficient trapping of atmospheric constituents¹². Furthermore, repeated
deforestation can reduce atmospheric P inputs to a level that induces a long-term negative
ecosystem P balance¹³. As such, the loss of this canopy trap through repeated clearing might
also reduce cumulative P inputs to an extent where some forests completely lose the capacity
to naturally recover from disturbances¹⁴. Understanding the biogeochemical and physical
interactions of canopy trapping and atmospheric P deposition along secondary forest
succession is vital for tropical landscapes, which face increasing anthropogenic pressures.
These effects are particularly relevant in Africa, where secondary forests are expected to
increasingly dominate the landscape as a result of ongoing slash-and-burn agricultural
practices^{15,16}. The African continent represents nearly 65% of the global burnt area annually¹⁷,
~~in part due to the prevalence of slash-and-burn agriculture^{18,19}—and associated high rates of~~

[revised manuscript text omitted]

~~and more complex canopies via intensified slash-and-burn and deforestation practices may~~
~~result in the overall reduction in atmospheric P inputs to the Congo basin forest floor.~~ Thus,
long-term absence of a canopy ~~disturbance , could result in negative ecosystem feedback,~~
~~where reduced input of limiting nutrients~~ might thereby enhance a long-term negative

ecosystem nutrient balance, ~~could and~~ constraint future forest recovery, ~~that should be~~
 ~~considered in land use management~~, much like as described by Lawrence and colleagues¹³.

 **Fig. 3.** Weekly rainfall amount (open field rainfall volume for agricultural fields, throughfall volumes for the
 forest stages) versus weekly phosphorus (P) deposition, in agricultural field (Ag), 5- (5yrs), 12- (12yrs), 20-
 (20yrs) and 60-year-old forest (60yrs). Solid lines show significant log-log fits via standardized major axis
 regression, while dashed lines indicate insignificant regressions (P-value>0.1). Boxplots show the median, the
 25th and 75th percentile and the 1.5 interquartile range spread for the weekly P deposition loads. Points in
 represent the individual weekly measurements of composite samples (n=8) per plot for rainfall volume and P
 deposition.

 **Forest canopies as temporary reservoirs for P deposition.** We used simulated black carbon
 surface mass ~~density concentration~~ (~~BCCMD~~~~BCSMC~~) from the MERRA2 model (Fig. 4dS2),
 to assess whether temporal P deposition patterns could be explained by temporal variations of
 black carbon in the atmosphere. However, we did not find a significant relationship between

[revised manuscript text omitted]

**Uncertainties.** This study reports some of the first *in situ* measured P deposition loads for central Africa.
However, due to logistical constraints and local conditions, we had to design the monitoring setup in a robust
and low-tech manner. These adaptations for the long-term monitoring might bring about some additional
uncertainties we want to report here. First: the rainfall and throughfall collectors we used in the field did not
have cooling (no grid power available, and safety issues in maintaining technological setups in the field), nor did
we add preservation chemicals to the collection bottles (due to the environmental risks involved with the
disposal of mercury, and the contamination of future analyses - thymol). Nevertheless, we judge that the
consequences of this are minor for total phosphorus concentration in the samples. ~~For~~ The DBC quantification,
by definition, was performed on filtered samples and thus: ~~as we filtered the samples prior to analysis, we~~ did not
include particulate BC (PBC) ~~in these analysis~~. However, we use DBC merely as a tracer for (dissolved) total P
sourced by biomass burning. Given the pre-dominance of biomass burning as the source of combustion derived
aerosols in the Congo (consistent source), ~~and~~ the long-range transport of aerosols to our study site, and the sub
0.2-micron size fraction of these aerosols, we are confident that ~~there is no reason to expect the proportions of~~
~~DBC and PBC to vary substantially through time~~ is a reliable tracer for combustion-derived inputs. Finally, we
use 'wet' and 'dry' deposition throughout this manuscript, while in fact our wet deposition was not collected
with 'wet-only' collectors. In practice, the open field depositional loads will also have dry deposition collecting
on the collector surface. Hence, our 'wet' deposition estimates are slightly overestimating wet deposition, while
the 'dry' deposition estimates are slightly underestimating actual dry deposition.

**Data availability**

The authors declare that all data supporting the findings of this study are available within the paper and its
supplementary information files.

**Acknowledgements**

We thank our collaborators at the Université de Kisangani and the Université Catholique de Bukavu for the local
expertise and support in setting up this research. M. Bauters is funded by the Fonds Wetenschappelijk
Onderzoek (FWO) Flanders, through a post-doctoral fellowship. We thank the people of the Yoko village for the
long-lasting collaboration and safeguarding of our equipment in the experimental forest. Some analyses and
visualizations used in this paper were produced with the Giovanni online data system, developed and maintained
by the NASA Goddard Earth Sciences Data and Information Services Center. We also acknowledge the MODIS
mission scientists and associated NASA personnel for the production of the data used in this research effort. H.
Verbeeck would like to acknowledge the financial support from the Fonds Wetenschappelijk Onderzoek (FWO;
grant no. G018319N). Support was also provided by the National Science Foundation OCE #1756812 to S.
Wagner. T. Drake was supported by the core funding of ETH Zurich granted to Johan Six. Analyses and
visualizations used in this paper were produced with the Giovanni online data system, developed and maintained
by the NASA GES DISC. We acknowledge the mission scientists and Principal Investigators who provided the
data used in this research effort.

**Author contributions**

404 M.Bauters, T.W.D and P.B. conceived the study. M. Bauters and K.V.O. carried out the drone flights. K.V.O.
processed the drone imagery for the 3D canopy reconstruction. M. Bauters, C.E. and I.M. oversaw and carried
out field work, and set up the permanent sampling units. M. Bauters, T.W.D., S.Bodé and S.W. analyzed the
dissolved black carbon in the composite samples. M. Bauters, I.M., P.B. and S.Bodé analyzed the phosphorus in
the sample set. I.M., C.E., L.C., M. Barthel and S. Baumgartner provided logistical support for the field work.
409 M.Bauters wrote the paper, with significant contributions from T.W.D., S.W., K.V., H.V., P.B. and K.V.O.

**Competing interests**

The authors declare no competing interests.

**Additional information**

Supplementary information is available for this paper at XXX.

Reprints and permissions information is available at XXX

Correspondence and requests for materials should be addressed to M. Bauters

[revised manuscript text omitted]

REVIEWERS' COMMENTS

Reviewer #1 (Remarks to the Author):

This manuscript has improved as the authors took into consideration the remarks put forward by the reviewers. There are still two points that need to be clarified before a possible publication of this work:

1) As indicated by reviewer 2, the rain collectors are not collecting only wet deposition but a quantity linked to total deposition. This should be more clearly indicated in the text since any later attempt for modellers to separate dry and wet deposition is not possible.

2) Line 163 the authors indicate that Figure 4d represents surface concentration of Black Carbon (BC) when the Figure as it stands shows Aerosol Optical Depth @ 550nm. The authors should either correct the text or update the Figure.

Reviewer #3 (Remarks to the Author):

In this revised version of the ms all the issues and concerns raised by the reviewers have been sufficiently addressed and the ms was improved. The point I raised on rugosity has been nicely incorporated. I agree with the authors response to Reviewer #2's methodological concerns, that these concerns have minor influence and do not compromise the results.